

# Improved Hilbert space exploration algorithms for finite temperature calculations

**Albertus J. J. M. de Klerk**[⋆] **and Jean-Sébastien Caux**[†]

Institute for Theoretical Physics, University of Amsterdam,
Postbus 94485, 1090 GL Amsterdam, The Netherlands

⋆ a.j.j.m.deklerk@uva.nl , † j.s.caux@uva.nl

## Abstract

Computing correlation functions in strongly-interacting quantum systems is one of the most important challenges of modern condensed matter theory, due to their importance in the description of many physical observables. Simultaneously, this challenge is one of the most difficult to address, due to the inapplicability of traditional perturbative methods or the few-body limitations of numerical approaches. For special cases, where the model is integrable, methods based on the Bethe Ansatz have succeeded in computing the spectrum and given us analytical expressions for the matrix elements of physically important operators. However, leveraging these results to compute correlation functions generally requires the numerical evaluation of summations over eigenstates. To perform these summations efficiently, Hilbert space exploration algorithms have been developed which has resulted most notably in the ABACUS library [1]. While this performs quite well for correlations on ground states or low-entropy states, the case of high entropy states (most importantly at finite temperatures or after a quantum quench) is more difficult, and leaves room for improvement. In this work, we develop a new Hilbert space exploration algorithm for the Lieb-Liniger model, specially tailored to optimize the computational order on finite-entropy states for correlations of density-related operators.

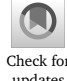
# 1   Introduction

The study of integrable models in one dimension has been a subject of interest ever since Bethe's solution of the Heisenberg model [2], but has gained broader attention in recent decades due to advances in both theory and experiment. On the theoretical side, Bethe Ansatz techniques have gone beyond the original spectrum-limited results and given us efficient expressions for matrix elements of physically important operators [3–7]. On the experimental side, advances in the field of ultracold atoms [8,9] allowed for the experimental realization of the Lieb-Liniger model and demonstrated the implications of integrability on non-equilibrium dynamics and equilibration [10]. Most importantly, integrable models have supplied a fertile ground to investigate the general principles governing out-of-equilibrium physics and helped improve our understanding of strongly correlated physics in one dimension [11–20]. The Lieb-Liniger model [21–24], perhaps the simplest non-trivial example of an integrable model in one dimension, describes a gas of one-dimensional bosons interacting via a delta function interaction potential and has received a lot of interest [25–53]. It is this model that we consider throughout this paper, although we comment on how the ideas in the paper can be extended to spin chains in Sec. 9.

The Algebraic Bethe Ansatz [5] gives us efficient expressions for important matrix elements, but even this combined with a knowledge of the spectrum is not enough to analytically calculate important correlation functions. Though there exist some partial analytical results (e.g. few-spinon contributions to zero-field correlations in spin-1/2 chains), in general one has to resort to explicit numerical summations over intermediate states in order to obtain quantitative results. Studies into these algorithms led to a set of highly efficient scanning algorithms encoded in ABACUS [1]. However, no universally optimal approach for such algorithms is known, making it interesting to investigate alternative approaches and see if we can improve upon the state of the art.

In this work, we develop novel scanning algorithms, inspired by those implemented in ABACUS, which we compare to the current version of ABACUS by considering the dynamical structure factor. We show that at zero temperature the performance of our algorithms is virtually identical to those in the current version of ABACUS, whereas we note that our algorithm performs more optimally for the finite temperature calculation. We also compare ABACUS to our algorithms for generating an optimal basis for computing local observables following a quench in the interaction strength [54]. In this case we notice a more dramatic increase in performance at finite temperature, highlighting the advantages of our approach for problems where states with multiple particle-hole excitations relative to the thermal state play an important role.

We start by reviewing the Lieb-Liniger model in Sec. 2, where we review the calculation of the DSF as well as the quench of interaction strength. This shows the need for Hilbert space scanning algorithms whose general principles we discuss in Sec. 3. We then start by

writing down the simplest algorithm that satisfies the general rules we require of all scanning algorithms in Sec. 4. This initial algorithm does not target a specific momentum sector, but with a simple addition to the rules of the algorithm it can be made to do so, as we show in 5. In Sec. 6 we show how this simple algorithm can be improved upon by adding additional rules relating to the preservation of the number of particle-hole pairs. In Sec. 7 we show how the algorithm can be improved even more not by adding additional rules, but by cleverly prioritizing certain parts of the calculation. The resulting algorithm is then compared to the existing state of the art in Sec. 8, where we show the advantages of our algorithm for finite temperature computations. We conclude and discuss how our insights are relevant to scanning algorithms for the spin chain in Sec. 9.

## 2  The Lieb-Liniger model

The Lieb-Liniger model [22,23] describes one-dimensional bosons with a delta function interaction potential, so its Hamiltonian can, in first quantised form, be written as

$$H = \sum_{i=1}^{N} \left[ -\frac{\hbar^2}{2m}\frac{\partial^2}{\partial x_i^2} + 2c\sum_{i<j}\delta(x_i - x_j) \right], \tag{1}$$

in the case where we have $N$ particles and $x_i$ is the position of the $i^{\text{th}}$ particle. Furthermore, $c$ represents the interaction strength and $m$ the mass. We take $\hbar = 1 = 2m$ to define our units. In the following we will often consider the second quantised form of this Hamiltonian, which with $\hbar$ and $2m$ set to one reads

$$H = \int_0^L dx \left[ \partial_x\Psi^\dagger(x)\partial_x\Psi(x) + c\Psi^\dagger(x)\Psi^\dagger(x)\Psi(x)\Psi(x) \right]. \tag{2}$$

Here $\Psi^\dagger$ is the bosonic creation operator which satisfies the canonical commutation relations,

$$\left[\Psi(x), \Psi^\dagger(y)\right] = \delta(x - y). \tag{3}$$

One of the interesting features of the model is that we can solve its Schrödinger equation to obtain the complete spectrum and all the corresponding eigenstates, making it a perfect candidate for studying features that are normally not accessible in interacting quantum many-body systems, such as the time evolution following a quantum quench. Furthermore, it turns out that the resulting time-evolution is different from those of generic quantum systems as found experimentally in cold atom experiments [10].

In the limit where the interaction strength vanishes we are dealing with free bosons so we can readily diagonalise the Hamiltonian using a Fourier transform. In the opposite (Tonks-Girardeau) limit where the bosons are infinitely repulsive, we can map the problem to that of non-interacting spinless fermions in one dimension [55]. In the intermediate regime the eigenstates can be found by considering what is known as the Bethe Ansatz, first used to solve the one-dimensional Heisenberg spin chain [2].

The Bethe Ansatz approach to solving the Lieb-Liniger model begins by considering the fundamental domain $D_N$, which for $N$ particles is defined as

$$D_N = \{x \in \mathbb{R}^N \mid x_1 < x_2 < \cdots < x_N\}. \tag{4}$$

Note that the restriction of the problem to this domain does not constitute a loss of generality as we can extend this solution to $\mathbb{R}^N$ by invoking the symmetry requirements of the wavefunction when exchanging particles.

The Bethe Ansatz is then that the wavefunction is a superposition of all permutations of plane waves with quasi-momenta $\lambda_j$ (often called rapidities) and amplitudes $A_\sigma$ that are at this stage undetermined giving

$$\Psi_N(x) = \sum_{\sigma \in S_N} A_\sigma e^{\sum_{j=1}^N i\lambda_{\sigma(j)} x_j}, \tag{5}$$

where $S_N$ is the group of permutations of the numbers $\{1, \ldots, N\}$. The coefficients $A_\sigma$ can then be determined by the boundary conditions for the fundamental domain arising from the delta function interaction potential for two-particle collisions. This gives

$$A_\sigma = (-1)^{[\sigma]} \prod_{1 \le l < j \le N} (\lambda_{\sigma(j)} - \lambda_{\sigma(l)} + ic), \tag{6}$$

where $(-1)^{[\sigma]}$ is the sign of the permutation. Imposing periodic boundary conditions on the Bethe wave function gives rise to what are called the Bethe equations, which determine the values of the rapidities $\lambda_j$, which read

$$e^{i\lambda_j L} = \prod_{l \ne j} \frac{\lambda_j - \lambda_l + ic}{\lambda_j - \lambda_l - ic}. \tag{7}$$

Any set of rapidities $\{\lambda_i\}_{1 \le i \le N}$ satisfying Eq. (7) thus gives rise to an eigenstate of the Lieb-Liniger model with energy $\sum_j \lambda_j^2$ and momentum $\sum_j \lambda_j$.

Instead of considering the Bethe equations directly, it is useful to take the logarithm of Eq. (7) resulting in the logarithmic Bethe equations

$$\lambda_j + \frac{2}{L} \sum_{k=1}^N \operatorname{atan}\left(\frac{\lambda_j - \lambda_k}{c}\right) - \frac{2\pi I_j}{L} = 0, \tag{8}$$

where we introduced the quantum numbers $\{I_j\}_{1 \le j \le N}$ which are integers when $N$ is odd and half-odd integers when $N$ is even. The quantum numbers are more than a mathematical necessity introduced by the logarithm, they turn out to be a convenient way of uniquely labelling the eigenstates. There is a one-to-one correspondence between the quantum numbers and the rapidities which respects the ordering, meaning that if $I_j > I_k$ then $\lambda_j > \lambda_k$ due to the monotonic nature of the second term on the left hand side of Eq.(8). Furthermore, the sum of the quantum numbers is proportional to the momentum of the eigenstate via

$$P = \sum_j \lambda_j = \frac{2\pi}{L} \sum_j I_j. \tag{9}$$

Finally, no two quantum numbers can be equal since the wavefunction then formally vanishes.

The solvability of the model is a result of the fact that all interactions can be reduced to two-body interactions, a hallmark of integrability. Another special property of the Lieb-Liniger model, also sometimes used to characterize integrability, is that it has infinitely many nontrivial commuting conserved charges whose eigenvalues are given by

$$Q_n = \sum_{j=1}^N \lambda_j^n, \tag{10}$$

for any $n \in \mathbb{N}$. Here $Q_1$ and $Q_2$ represent the momentum and energy respectively.

The ground state of the Lieb-Liniger model is the state whose quantum numbers are as close to zero as possible. Since the quantum numbers are not allowed to coincide and they

can only take on integer or half-odd integer values, this results in a configuration like a Fermi sea. To be precise, the quantum numbers of the ground state are given by

$$\left\{ -\frac{N+1}{2}, -\frac{N+1}{2}+1, \ldots, \frac{N-1}{2} \right\}. \tag{11}$$

We can therefore define a "Fermi momentum" in analogy with the Fermi sea, given by $k_F = \frac{\pi}{L}(N - \frac{1}{2})$ so that $k_F$ is between the last occupied and first unoccupied mode.

Taking the thermodynamic limit of the logarithmic Bethe equations leads to the thermodynamic Bethe Ansatz. In this limit the states are described by the continuum version of the quantum numbers called a root distribution. Root distributions corresponding to finite temperature states can be determined and discretized in order to obtain sets of quantum numbers at finite size that best resemble these distributions. These finite size approximations of the thermodynamic root distributions are called representative states [5].

Another important property of the Lieb-Liniger model is that efficient expressions for matrix elements of operators such as the density operator

$$\rho(x) = \Psi^\dagger(x)\Psi(x), \tag{12}$$

as well as the $g_2$ operator

$$g_2(x) = \Psi^\dagger(x)\Psi^\dagger(x)\Psi(x)\Psi(x), \tag{13}$$

have been obtained using Algebraic Bethe Ansatz techniques [6, 7, 56]. Knowledge of matrix elements of operators in combination with the solvability of the model is what allows us to study the correlation functions as well as the non-equilibrium time-evolution of observables in the Lieb-Liniger model. The time evolution of local observables in integrable models is of particular interest as, at long time after driving the system out of equilibrium, expectation values do not thermalize and instead relax to values determined from the Quench Action method [57–59], or (when applicable) a GGE [60–62]. This has not only been understood theoretically, but has been directly observed in experimental studies [10].

A property of the matrix elements crucial to the algorithms in this article, is that on average, the off-diagonal matrix elements are largest when the bra and ket states share the most quantum numbers.[1] To see why, note that Slavnov's formula for the overlap $\langle \mu | \lambda \rangle$ between a Bethe state $|\lambda\rangle$ and an arbitrary set of rapidities $|\mu\rangle$ has poles for coinciding rapidities [5]. As such, overlaps between Bethe states are maximal when the number of (close to) coinciding rapidities is maximal. The matrix elements of the density and $g_2$ operators are derived from this formula for the overlaps by determining the action of the operator under consideration on the bra or ket and using the overlap formula. For example, acting with $\Psi(0)$ on a Bethe state results in a superposition of states where one of the rapidities is removed and the others are shifted due to the interactions. Therefore, the off-diagonal matrix elements of the density operator are maximal when the bra and ket differ by one quantum number. As the number of differing quantum numbers increases, the matrix element becomes smaller due to the smaller number of (close to) coinciding rapidities. Similarly, the off-diagonal matrix elements of the $g_2$ operator are maximal when the bra and ket differ by one or two quantum numbers. The importance of the number of differences between the quantum numbers of the bra and ket is also influenced by the interaction strength, with its importance diminishing as the interaction strength decreases.

---

[1]Here it is good to note that how much the quantum they do not share are different is of secondary importance. For example, even when one of the quantum numbers $I_j$ and the corresponding rapidity $\lambda_j$ go off to $\pm\infty$, the term in the logarithmic Bethe equations, corresponding to the rapidity at infinity evaluates to $\pm\pi/L$ which effectively shifts $I_k$ by $\mp1/2$. Thus even a massive change in one quantum number shifts the other quantum numbers only a little.

To see how the need for Hilbert space scanning algorithms arises, consider the computation of the dynamical structure factor in the ground state of the Lieb-Liniger model, given by

$$S(k,\omega) = \int_0^L dx \int dt\, e^{-ikx+i\omega t} \langle \rho(x,t)\rho(0,0)\rangle = \frac{2\pi}{L}\sum_\alpha |\langle 0|\rho_k|\alpha\rangle|^2 \delta(\omega - E_\alpha - E_0), \quad (14)$$

where $\rho_k$ is the Fourier transform of the density operator. In order to numerically approximate Eq. (14), we need a set of eigenstates $|\alpha\rangle$ for which we compute the energies $E_\alpha$, the matrix elements $\langle 0|\rho_k|\alpha\rangle$, and perform the summation. As the Lieb-Liniger model possesses an infinite number of eigenstates, an evaluation of the sum in Eq. (15) neccessitates a truncation and the accuracy of the calculation depends on the number of eigenstates and their matrix elements $\langle 0|\rho_k|\alpha\rangle$. The convergence is quantified by the $f$ sum rule [63], which states that

$$\int \frac{d\omega}{2\pi} \omega S(k,\omega) = \frac{Nk^2}{L}\,. \quad (15)$$

Given Eq. (15), we can convert the contributions to the summation in Eq. (14) into a weighing function for a given eigenstate $|\alpha\rangle$ given by

$$w_f(\alpha) = \frac{L}{Nk^2}(E_\alpha - E_0)|\langle 0|\rho_k|\alpha\rangle|^2\,, \quad (16)$$

for $k \neq 0$ such that the summation over the weights of all eigenstates $|\alpha\rangle$ at a fixed momentum value gives 1. When considering the summation over the weights of Eq. (16), there are two contributing factors that determine the importance of an eigenstate, its energy and the matrix element. For the Lieb-Liniger model it turns out that most matrix elements except for a tiny portion of states are vanishingly small, dominating the effect the energy has. Therefore our focus lies primarily on finding those states for which the matrix elements are large in order to get a good saturation of the sumrule. Hilbert space scanning algorithms are designed to preferentially generate eigenstates for which for example $w(|\alpha\rangle) = |\langle 0|\rho_k|\alpha\rangle|^2$ or $w(|\alpha\rangle) = (E_\alpha - E_0)|\langle 0|\rho_k|\alpha\rangle|^2$ are maximal [1, 25, 29].

Another problem where the need for generating appropriate eigenstates arises, is when choosing a basis for truncated spectrum methods [64–66]. Consider a quench of the Lieb-Liniger model where we change the interaction strength at $t = 0$ from $c_i$ to $c_f$. Truncated spectrum methods can be used to compute the time evolution of the initial state $|\Psi_0\rangle$ in terms of a set of eigenstates of the Lieb-Liniger model at interaction strength $c_f$, i.e.

$$|\Psi_0(t)\rangle = \sum_\alpha b_\alpha e^{-iE_\alpha t}|\alpha\rangle\,, \quad (17)$$

where $b_\alpha = \langle \alpha|\Psi_0\rangle$ are the coefficients being approximated by the truncated spectrum methods. However, the accuracy of the expansion in Eq. (17) depends on the choice of basis states $|\alpha\rangle$. By choosing a weighing function that approximates $\langle \alpha|\Psi_0\rangle$ we can leverage our scanning algorithms to generate a close to ideal basis for this quench [67].

## 3  Scanning Algorithms

The algorithms for exploring Hilbert space that we present in this article can all be generally described as generating a single-rooted tree where each node represents an eigenstate. The algorithms differ in the rules that determine which new eigenstates are generated from a given node, resulting in trees with different topologies even when they are generated from the same initial eigenstate, which we will herein call the seed state. In some cases, the seed

state for an algorithm will be directly related to the observable we are trying to compute (for example, for the calculation of the dynamical structure factor, Eq. (14), the seed state will be the ground state) while for other problems it may not be (this is the case for the quench problem, which we discuss later). This approach using tree-building algorithms was shown to be very successful for the computation of correlation functions in integrable models [1]. The purpose of this article is to introduce new algorithms for scanning and comparing their properties for different problems.

There are two properties we require of all of our Hilbert space exploration algorithms:

1. *Uniqueness:* No eigenstate should come up more than once when generating a tree of eigenstates.

2. *Completeness*: All eigenstates in a pre-defined sector of Hilbert space must occur in the tree if we give the algorithm infinite computation time.

Property 1 ensures that we can use the eigenstates generated in a tree to perform a summation such as the ones in Eq. (14) and Eq. (17) without having to separately keep track of which eigenstates we already included. Property 2 comes from the fact that we want to be able to approximate summations such as the one in Eq. (14) arbitrarily well given infinite computational resources. If some states were not generated by the algorithm, this would not be possible.

In the absence of a UV cut-off, any momentum sector of the Lieb-Liniger model is infinite dimensional even in a finite volume system. Therefore we can never truly generate the full corresponding tree with finite computational resources. As such, it is not only relevant what the final tree would look like, but also how it is built, i.e. what it looks like after some finite time. In order to ensure that we spend our time wisely, we can pause the generation of new eigenstates from certain branches of the tree. The goal here is to pause the generation of eigenstates in the algorithm until they are the most important ungenerated eigenstates that are remaining. The result is that after some finite time we end up with a truncated tree where most of the nodes on the outside of the tree could be used to generate additional new eigenstates.

The way in which we determine which branches to pause at a given time and the interplay between this and the rules for growing the tree as determined by our algorithm can have a strong influence on the way the tree is grown and therefore the quality of our algorithm. To see this, consider an algorithm that satisfies the completeness and overcounting criteria. Now suppose we run this algorithm for some time which gives rise to a growing tree as depicted in Fig. 1. The white nodes represent eigenstates whose weight is below some threshold value whereas the weight of the grey nodes is bigger, allowing us to visually distinguish between high-weight and low-weight nodes. The algorithm showcased here is not ideal since the high-weight nodes 7 and 8 are generated after the low-weight nodes 5 and 6. When scaled up this means we can, at finite runtime, miss important states because they are effectively locked behind low-weight states. In an ideal algorithm the descendents of a node would therefore always have a lower weight than the parent node.

The extent to which the descendents of nodes have a weight lower than their parents depends on an interplay between the rules of the algorithm and the weighing function considered. Therefore there may be some algorithms which are more compatible with certain weighing functions and others that are more compatible with others. For the Lieb-Liniger model, which we consider in this article, we can identify a criterion that is common to the weighing functions we want to consider. This criterion is the number of particle-hole excitations as it is closely related to the matrix element value of the operators we want to consider as explained in Sec. 2. This allows us to create close to optimal algorithms for this system.

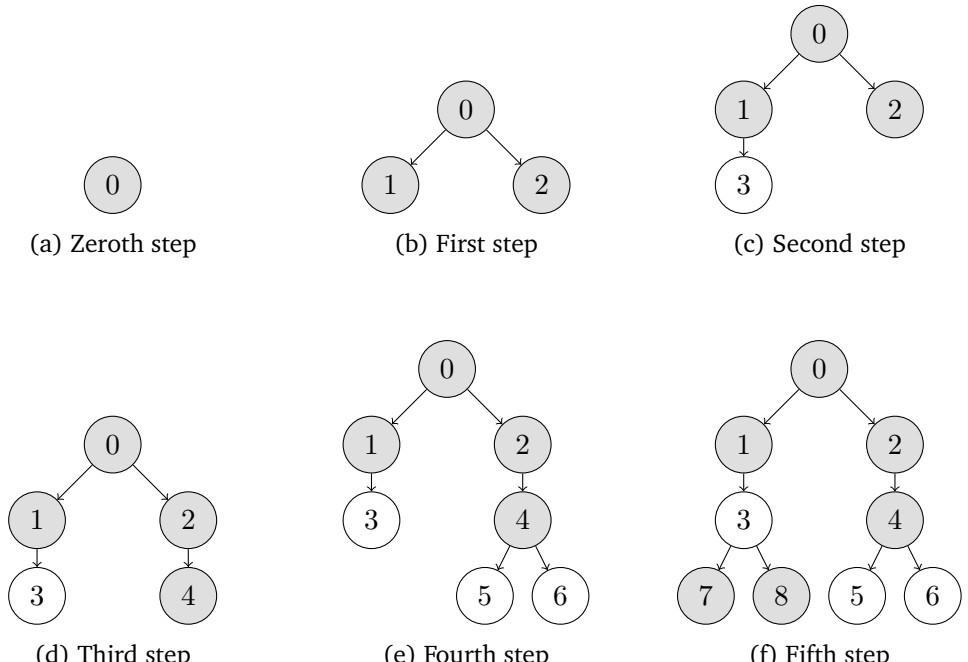

Figure 1: Illustration of the state of the tree after the first six steps of a non-optimal algorithm. The white circles represent eigenstates whose weight is smaller than some threshold value, whereas the grey circles represent eigenstates whose weight is larger than the same threshhold. An optimal algorithm would therefore not generate grey circles after white circles, which we see occuring in the fifth step of the algorithm.

## 4 A basic scanning algorithm

In order to introduce a way of visualising the quantum numbers, consider a set of quantum numbers $\{I_i\}_{i\leq 5}$ which we assume to be ordered, an assumption we retain for the remainder of this article. Since no two quantum numbers can be the same, we can think of them as five particles on a one-dimensional lattice, which can be visualized as follows:

$$\cdots \; \circ \; \bullet \; \circ \; \bullet \; \bullet \; \bullet \; \circ \; \bullet \; \circ \; \cdots$$
$$\text{-4 -3 -2 -1 0 1 2 3 4}$$

The numbers below the circles represent the number corresponding to the position on the line, so this state represents $\{-3, -1, 0, 1, 3\}$. In this article we visualize the quantum numbers in order to illustrate the principles of our algorithms. For this purpose we are not interested in the absolute values of the quantum numbers, but rather only the differences between the quantum numbers allowing us to drop the numbers below the circles going forward.

One of the simplest ways of generating a new set of quantum numbers from a given set is by changing one of the quantum numbers by ±1, the minimal possible value provided that this does not render two quantum numbers equal. Such a change can be identified with a particle hopping to the left or right on the lattice, where a particle hopping to the right looks like

$$\cdots \; \circ \; \bullet \; \circ \; \bullet \; \bullet \; \bullet \; \circ \; \bullet \; \circ \; \cdots$$
$$\downarrow$$
$$\cdots \; \circ \; \circ \; \bullet \; \bullet \; \bullet \; \bullet \; \circ \; \bullet \; \circ \; \cdots$$

and a particle hopping to the left looks like

$$\cdots \; \circ \; \bullet \; \circ \; \bullet \; \bullet \; \bullet \; \circ \; \bullet \; \circ \; \cdots$$
$$\downarrow$$
$$\cdots \; \bullet \; \circ \; \circ \; \bullet \; \bullet \; \bullet \; \circ \; \bullet \; \circ \; \cdots$$

In both cases the first line represents the quantum numbers of the initial state and the line following that represents the quantum numbers of its descendent where one coloured particle has hopped to a neighbouring lattice site.

Such particle hops can be used to formulate rules for generating descendents and building a tree, but we need to impose rules to avoid overcounting. We start with an algorithm where the descendents of a node are those where a single particle has hopped one position to the right or left provided that this does not result in a collision of particles. In the following we identify the issues with this algorithm and propose solutions, the result of which will be our first real scanning algorithm which does not overcount states.

The first issue we consider is that when a particle first moves to the right and then back it will result in the same state we started with. Without any restrictions we can generate subtrees that look like

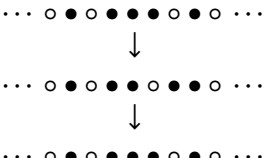

The first rule of our scanning algorithms is therefore that once a particle has moved to either the left or the right, it can only continue moving in that direction. We refer to particles that have moved to the right as rightmovers whereas we refer to particles that have moved to the left as leftmovers. In our visualizations we colour the leftmovers blue and the rightmovers orange.

The second issue is due to there being no preferred ordering of moving particles to the left or right, as with the current rules the following subtree could be generated

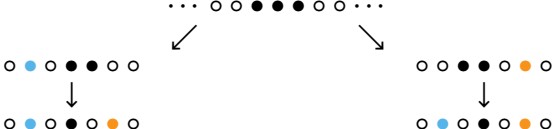

Here we see that the bottom two states are the same even though they are not the same node in the tree and no particle has changed direction. In order to avoid overcounting we impose the rule that if a state has a rightmover then its descendents can not have additional leftmovers. In the subtree we just considered this means that the state on the bottom right would not have been generated as there is already a rightmover (the right most particle in the above figure).

The final issue that arises is similar to the previous one, only now it involves only moves to the left or right. To understand the issue, note that with our current rules the following subtree could be generated

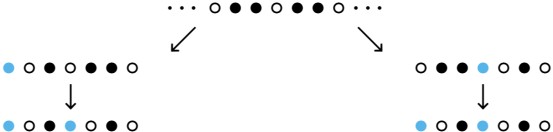

Again, we see that the two lowest configurations of integers are identical, despite being on different branches of the tree. This issue can be avoided by only allowing particles to hop to the left if they are the rightmost leftmover or to its right. This would rule out the configuration on the bottom right, as it is generated by creating a leftmover to the left of the existing leftmover.



We can run into the same problem when considering rightmovers giving rise to

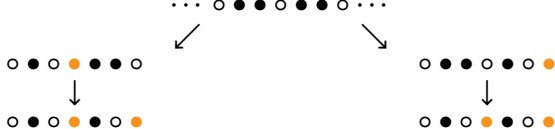

which can be avoided by only allowing particles to hop to the right if they are the leftmost rightmover or to its left.

The rules we have imposed thus far constitute the first scanning algorithm that is complete and does not count the same state twice as we will show shortly. For now, let us summarise the rules of the algorithm.

Let $\{I_i\}_{i \leq N}$ be the set of quantum numbers at a node. We denote the leftmoving and rightmoving quantum numbers by $\{I_j^L\}_{j \leq N_L}$ and $\{I_j^R\}_{j \leq N_R}$ respectively.

- **Move quantum numbers to the right:** Generate a descendent for every $I_l \in \{I_i\}_{i \leq N}$ for which $I_l \notin \{I_j^L\}_{j \leq N}$, $I_l \leq I_0^R$, and $I_l + 1 \neq I_{l+1}$ if $l \neq N$ with quantum numbers given by $\{I_i + \delta_{l,i}\}_{i \leq N}$.

- **Move quantum numbers to the left:** If $\{I_k^R\} = \emptyset$ generate a descendent for every $I_l \in \{I_i\}_{i \leq N}$ such that $I_l \geq I_{N_L}^L$, and $I_{l-1} \neq I_l - 1$ if $l \neq 0$ with quantum numbers given by $\{I_i - \delta_{l,i}\}_{i \leq N}$.

The maximal number of descendents is therefore $2N$. What remains is the proof that this algorithm generates a tree containing every set of allowed quantum numbers exactly once regardless of the seed state chosen.

Consider an arbitrary seed state $\{I_i^{SS}\}_{i \leq N}$ and an arbitrary target state $\{\tilde{I}_i\}_{i \leq N}$. In order to show completeness and the absence of overcounting it is sufficient to show that the target state occurs in the generated tree precisely once. This is equivalent to there being one way of applying the rules of the algorithm to get from the seed state to the target state. The quantum numbers of the node we consider at an intermediate step of the algorithm will be denoted by $\{I_i\}_{i \leq N}$.

Since our algorithm does not allow particles to hop to the left in the presence of rightmovers, we first have to move all particles for which $\tilde{I}_l < I_l^{SS}$. We first consider the leftmost particle of this type with index $l$ and claim that we can move it all the way to its target position without collisions. To show this, assume the contrary, which implies that $l \neq 1$ and $I_{l-1}^{SS} \geq \tilde{I}_l$. However, since $l$ was the index of the leftmost particle that had to move to the left, we know that $\tilde{I}_{l-1} \geq I_{l-1}^{SS}$. Combining these statements gives $\tilde{I}_{l-1} \geq I_{l-1}^{SS} \geq \tilde{I}_l$ which is a contradiction. Repeating this argument for all quantum numbers of the seed state that have to be decreased to reach the target state starting with the smallest one, we can move all such quantum numbers to the right position in a unique way.

Having fixed the quantum numbers that have to be decreased, we are left with quantum numbers for which $\tilde{I}_l \geq I_l^{SS}$. This time we claim that we can start from the rightmost particle for which this inequality holds and the rules of our algorithm allow it to be put in its place without collisions. To prove this, we again assume the contrary which implies that $l \neq N$ and that $I_{l+1} \leq \tilde{I}_l$. However, since $I_l$ was the rightmost quantum number that had to be increased, and since we already fixed the leftmovers we have that $I_{l+1} = \tilde{I}_{l+1}$. Together this gives $\tilde{I}_{l+1} = I_{l+1} \leq \tilde{I}_l$ giving the contradiction we require. This finishes the proof of completeness as well as showing that there is no overcounting.

Throughout the remainder of this article, we refer to the scanning algorithm developed in this section as stepwise scanning (SWS).

# 5 Imposing momentum conservation

In the previous section we introduced a scanning algorithm that would generate every state in Hilbert space exactly once, given infinite computational resources. However, often we are interested in a particular momentum sector, so for such problems it is not particularly well-suited. After all, it would mean that we are only interested in a tiny subset of the states that we generate. In this section we introduce a variant of the previous algorithm which restricts itself to a given momentum sector.

To turn stepwise scanning into an algorithm that generates descendents whose momentum is equal to that of their parent, we combine the rules we have for moving particles to the left and right. The momentum of a state is proportional to the sum of the quantum numbers, Eq. (9), so moving one particle one step to the right and another one step to the left ensures that momentum remains preserved. Furthermore, this approach eliminates the second problem we encountered in the previous section due to which we imposed the rule that no particle can hop left in the presence of rightmovers. The resulting algorithm goes as follows.

Let $\{I_i\}_{i \leq N}$ be the set of quantum numbers at a node. We denote the leftmoving and rightmoving quantum numbers by $\{I_j^L\}_{j \leq N_L}$ and $\{I_j^R\}_{j \leq N_R}$ respectively.

- **Generate rightmovers:** Generate an intermediate descendent $C_r$ for every $I_l \in \{I_i\}_{i \leq N}$ such that $I_l \notin \{I_j^L\}_{j \leq N_L}$, $I_l \leq I_0^R$, and $I_l + 1 \neq I_{l+1}$ if $l \neq N$.

- **Generate leftmovers:** For every intermediate descendent $C_r$ generate a descendent for every $I_l \in \{I_i^{C_r}\}_{i \leq N}$ such that $I_l \notin \{I_j^{C_r,R}\}_{j \leq N_{C_r,R}}$, $I_l \geq I_{N_{C_r,L}}^L$, and $I_l - 1 \neq I_{l-1}$ if $l \neq 0$.

The scanning routine described here, which we call momentum preserving stepwise scanning (SWS-MP), generates at most $N^2$ descendents in contrast to the at most $2N$ descendents in regular stepwise scanning. The difference arises due to the fact that we imposed momentum conservation, which led us to essentially apply first the first step of regular stepwise scanning, generating at most $N$ intermediate descendents, and then applying the second step of the stepwise scanning algorithm to these intermediate descendents.

To show that this algorithm is also complete in the sense that it can generate any state whose momentum is equal to that of the seed state, consider a random seed state $\{I_i^{SS}\}_{i \leq N}$ and a random target state $\{\tilde{I}_{i \leq N}\}$ with the same momentum. There are indices $k \in \{k_1, \ldots, k_{N_L}\}$ such that $\tilde{I}_k \leq I_k^{SS}$ as well as indices $l \in \{l_1, \ldots, l_{N_R}\}$ such that $\tilde{I}_l \geq I_l^{SS}$ which represent the indices of what will become the leftmovers and rightmovers respectively. Note that the size of these sets of indices can be different, as momentum preservation only requires the number of hops to the right and left to be preserved. In order to reach the target state from the seed state we again have to start by moving the leftmost leftmover, i.e. $I_{k_1}$, and the rightmost rightmover, i.e. $I_{N_R}$. The proof that these particles can hop to their target positions without collisions is exactly the same as the proof of completeness for regular stepwise scanning, so for that we refer the reader to the previous section. The same argument holds for the rightmovers being able to move to their target positions. Since there is still a unique order of moves by which we can reach the target state given the seed state, we also have no overcounting.

To understand the problems that arise when we try to apply this algorithm to situations where the momentum of the states we want to scan for is not equal to the natural candidate for the seed state, consider the example of the dynamical structure factor, see Eq. (14). Here the reference state we are interested in is the ground state $|0\rangle$ and we want to use a scanning algorithm to find the states $|\alpha\rangle$ at some fixed value of momentum most important to the summation. However, since the states in the intermediate summation do not belong to the same momentum sector as the ground state we cannot use a momentum preserving stepwise scanning algorithm in order to find them. In principle we can choose some other state from the

targeted momentum sector and use it as a seed state to the algorithm, but this raises the question of which state to choose. Even though in the limit where we have infinite computational resources this does not matter, it does affect what the tree looks like after a finite amount of time. After all, a bad choice can lead to the situation where important contributions are only generated after a long time because they are far down in the tree or are "hidden" as descendents of unimportant states. In fact, it is unclear if there even exists a seed state that would not lead to a Bethe tree with undesirable properties in this case.

In order to avoid having to choose a seed state for the target momentum sector, we choose to tweak the rules of the scanning algorithm such that we can use a given reference state (in the case of the dynamical structure factor, the ground state) despite its momentum not being equal to that of the momentum sector we are interested in. After all, we want to generate the states with few particle-hole pairs first as these are the states we generally expect to have the largest matrix elements for local operators. Taking the reference state to be the seed state ensures that these few particle-hole states are generated because the rules of our algorithm generate descendents with at most two additional particle-hole pairs. In order to generate states in the targeted momentum sector from a given reference state we add the rule that if the momentum of a node is smaller than the target momentum, its descendents are generated by following the first step of the algorithm, whereas if it is larger the rules of the second step are used. Note that applying only one of the two steps of momentum preserving stepwise scanning generates descendents whose momentum is changed by the minimal amount compared to their parent state. As such, there can be no overshooting of the target momentum sector. Furthermore, every branch created from the seed state reaches the target momentum sector in $\frac{k_{target}L}{2\pi}$ steps or dies off before then.

In order to assess the quality of this algorithm, we consider again the dynamical structure factor introduced in Eq. (14). Starting from the ground state, we can generate a tree of eigenstates where we use the weighing function that selects for states which contribute most strongly for the saturation of the $f$ sum rule as defined in Eq. (15). For now, we generate descendents node by node starting each time with one of the highest weight nodes as we elaborate more in section 7. The results for this calculation for three different values of the interaction strength are shown in Fig. 2a, b, c We see that in all three cases we reach close to optimal convergence with very few states meaning the most important states are generated first.

At finite temperature, we can do the same calculation provided that we replace the ground state with the representative state of the thermal state we wish to consider [5]. The calculation of the dynamical structure factor at finite temperature is inherently more difficult, but it also turns out that the simple scanning algorithm we have developed is not optimal for the finite temperature case. To understand why, consider Fig. 2d, e, f where we again consider the $f$ sumrule convergence. Besides the significantly poorer rate of convergence, we observe clear plateaus where convergence stagnates. This indicates that the algorithm is not succesfully generating the states that contribute to the finite-temperature correlation function most strongly first. In the next section we propose changes to our algorithm that lead to better convergence.

# 6 Imposing additional constraints

Thus far we have proposed algorithms where we generate descendents by moving quantum numbers by the minimal amount at each step of the algorithm, assuming this minimal change would result in the most important states first being generated first. However, it turns out that there is a property of the quantum numbers defining an eigenstate which is an important indicator of its importance for most of the weighing functions we are interested in that we

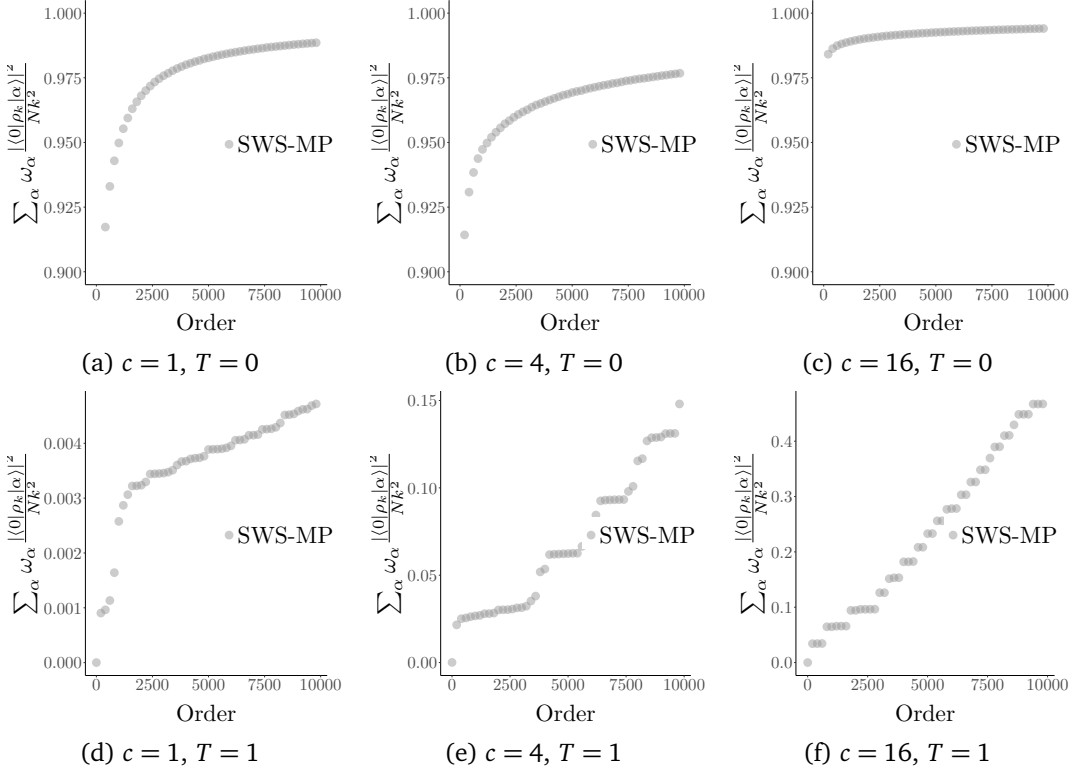

Figure 2: Saturation of the $f$ sumrule with the number of states included in the summation. Starting from the ground state for $(a)$-$(c)$ and the representative thermal state at $T = 1$ for $(d)$-$(f)$, we generated 10,000 states using momentum preserving stepwise scanning for a target momentum of $k = \pi$, and $N = 128 = L$. We plot the sum rule saturation after every 200 states for $c = 1$ in $(a)$ and $(d)$, for $c = 4$ in $(b)$ and $(e)$, and for $c = 16$ in $(c)$ and $(f)$. Convergence is near perfect after very few states in the zero temperature case, whereas in the finite temperature case convergence is poor for the number of states considered. Furthermore, the interaction strength is seen to be an important variable in the finite temperature case, with smaller interaction strengths corresponding to poorer convergence.

have ignored thus far. This property is the number of particle-hole pairs of a state with respect to a given state of importance to the calculation at hand, herein the reference state. In this section we explain how these particle-hole pairs are defined and we present algorithms that use knowledge of this property to produce more efficient algorithms.

In order to define the concept of particles and holes in this context, consider a seed state $\{I^{SS}_i\}_{i \leq N}$, visualised by

$$\circ\ \circ\ \bullet\ \bullet\ \bullet\ \bullet\ \bullet\ \circ\ \circ$$

Then for another state $\{I_i\}_{i \leq N}$, the quantum numbers $I_l$ which do not occur in $\{I^{SS}_i\}_{i \leq N}$ are called particles whereas the lattice positions which are now empty as a result are called holes. For example, for the following three states the particles and holes are labelled with a $p$ and $h$ respectively

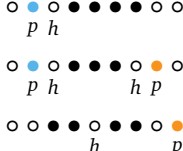

Note that we always have an equal number of particles and holes enabling us to talk about particle-hole pairs.

What makes the number of particle-hole pairs an important property to consider is that generally the fewer particle-hole pairs a state has, the bigger the off-diagonal matrix element between it and the reference state for local operators. This means that the states with few particle-hole pairs are generally those with the largest weights, meaning we should generate them first. For the regular and momentum preserving stepwise scanning algorithms, however, a descendent can have fewer particle-hole pairs than its parent. For example, consider a seed state given by

$$\circ \, \bullet \, \circ \, \bullet \, \circ \, \bullet \, \circ \, \circ \, \circ$$

then it generates the following subtree where a particle and hole annihilate one another

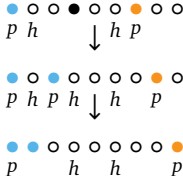

Taking on board that matrix elements of local operators depend upon the number of particle-hole excitations in a structured way, we can modify the previously proposed algorithms to more efficiently explore the Hilbert space.

One way to proceed is to simply forbid the annihilation of particle-hole pairs within a modified algorithm. However, this approach comes at a cost since adding this restriction to the rules of the regular and momentum preserving stepwise scanning algorithms breaks completeness. In order to regain this necessary feature, we allow quantum numbers that have not moved yet to hop into the position of a hole if it lies between it and one of the neighbouring quantum numbers. For example, this allows the following subtree to be generated:

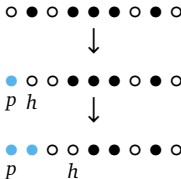

Note that this means that this allows for quantum number jumps of more than one position. Applying these changes to stepwise scanning scanning leads to the following rules, which constitute the leapwise scanning algorithm (LWS).

Let $\{I_i\}_{i \leq N}$ be the set of quantum numbers at a node. We again denote the leftmoving and rightmoving quantum numbers by $\{I_j^L\}_{j \leq N_L}$ and $\{I_j^R\}_{j \leq N_R}$ respectively. Furthermore, we label the positions of the holes as $\{I_j^h\}_{j \leq N_h}$ and the particles by $\{I_j^p\}_{j \leq N_p}$.

- **Generate higher momentum descendents:** Generate a descendent for every $I_l \in \{I_i\}_{i \leq N}$ for which $I_l \notin \{I_j^L\}_{j \leq N_L}$, $I_l \leq I_0^R$, and either

  - $I_l + 1 \neq I_{l+1}$ if $l \neq N$ with quantum numbers given by $\{I_i + \delta_{l,i}\}_{i \leq N}$, or
  - there exists a $k$ such that $I_l < I_k^h < I_{l+1}$ with quantum numbers given by $\{I_i + \delta_{i,l}(I_k^h - I_i)\}_{i \leq N}$.

- **Generate lower momentum descendents:** If $\{I_k^R\} = \emptyset$ generate a descendent for every $I_l \in \{I_i\}_{i \leq N}$ such that

○ $I_l \geq I_{N_L}^L$, $(I_l - 1) \notin \{I_j^h\}_{j \leq N_h}$, and $I_{l-1} \neq I_l - 1$ if $l \neq 0$ with quantum numbers given by $\{I_i - \delta_{l,i}\}_{i \leq N}$

○ there exists a $k$ such that $I_{l-1} < I_k^h < I_l$ with quantum numbers given by $\{I_i + \delta_{i,l}(I_k^h - I_i)\}$.

The proof that this algorithm is complete and does not overcount carries over directly from the proof for the stepwise scanning algorithm.

Like stepwise scanning, leapwise scanning is not suitable for targeting a fixed momentum sector. One step towards a solution of this problem is to, like before, combine the first and second step of the leapwise scanning algorithm. However, in the current case the resulting algorithm does not preserve momentum since the momentum increasing move from the first step and the momentum decreasing momentum from the second step may not cancel. The solution to this issue is the same as the solution to the problem we had with the momentum preserving stepwise scanning algorithm when we wanted to consider a target momentum sector whose momentum was different from that of the reference state. We generate the momentum increasing descendents for states whose momentum is smaller than the target momentum and momentum decreasing descendents for states whose momentum is larger. For states at the right momentum we combine the steps as we do in momentum preserving stepwise scanning. This algorithm, which we refer to as momentum preserving leapwise scanning (LWS-MP), can be summarised as follows.

Let $\{I_i\}_{i \leq N}$ be the set of quantum numbers at a node. We denote the leftmoving and rightmoving quantum numbers by $\{I_j^L\}_{j \leq N_L}$ and $\{I_j^R\}_{j \leq N_R}$ respectively. Furthermore, we label the positions of the holes as $\{I_j^h\}_{j \leq N_h}$ and the particles by $\{I_j^p\}_{j \leq N_p}$.

- **Generate rightmovers:** Generate an intermediate descendent $C_r$ for every $I_l \in \{I_i\}_{i \leq N}$ such that $I_l \notin \{I_j^L\}_{j \leq N}$, $I_l \leq I_0^R$ and either

  ○ $I_l + 1 \notin \{I_j^h\}_{j \leq N_h}$, and $I_l + 1 \neq I_{l+1}$ if $l \neq N$ with quantum numbers given by $\{I_i + \delta_{l,i}\}_{i \leq N}$, or

  ○ there exists a $k$ such that $I_l < I_k^h < I_{l+1}$ with quantum numbers given by $\{I_i + \delta_{i,l}(I_k^h - I_i)\}_{i \leq N}$.

- **Generate leftmovers:** For every intermediate descendent $C_r$ generate a descendent for every $I_l \in \{I_i^{C_r}\}_{i \leq N}$ such that $I_l \notin \{I_j^R\}_{j \leq N_R}$, $I_l \geq I_{N_L}^L$, and either

  ○ $(I_l - 1) \notin \{I_j^h\}_{j \leq N_h}$, and $I_{l-1} \neq I_l - 1$ if $l \neq 0$ with quantum numbers given by $\{I_i - \delta_{l,i}\}_{i \leq N}$, or

  ○ there exists a $k$ such that $I_{l-1} < I_k^h < I_l$ with quantum numbers given by $\{I_i + \delta_{i,l}(I_k^h - I_i)\}$.

If the momentum of the node is smaller than the target momentum, we only do the first step and the intermediate descendents become the descendents. On the other hand, if the momentum of the node is smaller than the target momentum we apply the second step to the node. The proof that the resulting algorithm is complete and does not overcount is the same as the proof for the momentum preserving stepwise scanning algorithm.

At zero temperature momentum preserving leapwise scanning gives practically the same results as momentum preserving stepwise scanning as can be seen in Fig. 3a, b, c. To understand this, note that at $T = 0$ we start from the ground state which consists of a single block of neighbouring quantum numbers without holes. The only initial excitations that are then allowed by our rules are ones that move the outermost quantum numbers out leaving behind holes. This in turn gives space for more particle-hole pairs to be created, but no jumps over

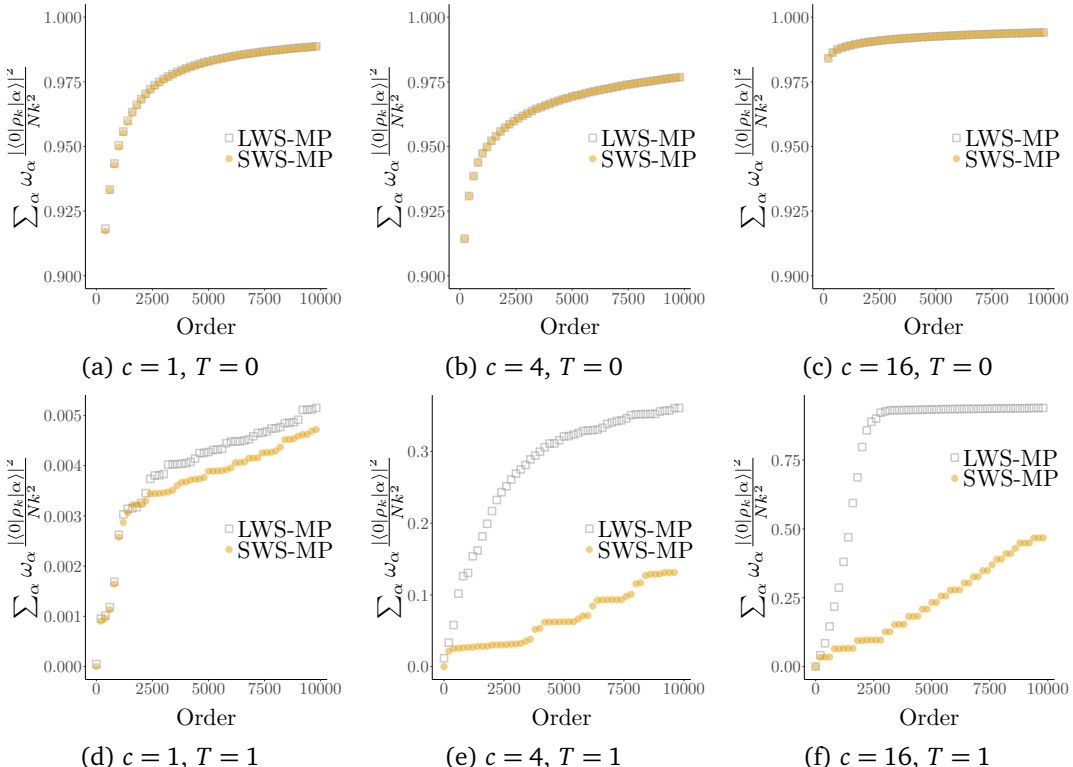

Figure 3: Comparison of the saturation of the $f$ sumrule with the number of states included in the summation between momentum preserving stepwise (SWS-MP) and momentum preserving leapwise (LWS-MP) scanning. Starting from the ground state for ($a$)-($c$) and the representative thermal state at $T = 1$ for ($d$)-($f$), we generate 10,000 states for a target momentum of $k = \pi$, and $N = 128 = L$. We plot the sum rule saturation after every 200 states for $c = 1$ in ($a$) and ($d$), for $c = 4$ in ($b$) and ($e$), and for $c = 16$ in ($c$) and ($f$). Convergence is identical for both momentum preserving stepwise and leapwise scanning zero temperature, but not for the finite temperature case where momentum preserving leapwise scanning performs better. The plateaus at large and intermediate interaction strengths are barely noticeable for momentum preserving leapwise scanning whereas they remain pronounced in the case where $c = 1$.

vacant positions that are not holes will occur during momentum preserving leapwise scanning. As a result, the descendents generated by momentum preserving stepwise and leapwise scanning as well as the resulting trees are virtually the same.

At finite temperature momentum preserving leapwise scanning outperforms momentum preserving stepwise scanning as can be seen in part Fig. 3d, e, f. In this case, we start from a seed state where there is no longer a zero temperature Fermi sea, but instead there are vacant positions between the quantum numbers, i.e. the quantum numbers of the seed state are no longer of the form $\{a, a + 1, a + 2, \dots\}$ but rather something like $\{a, a + 3, a + 4, a + 7, \dots\}$ for example. Therefore, when descendents are generated where quantum numbers neighbouring vacant positions are moved, they leave behind holes surrounded by vacant positions. For momentum preserving leapwise scanning, states with the same number of particle-hole excitations can be created in such a scenario by letting another quantum number jump over the vacant positions to this newly created hole position, whereas in the case of momentum preserving stepwise scanning a bunch of intermediate states would have to be created. Not

having to generate these intermediate states is what leads to the increase in efficiency.

# 7 Beyond the topology of the tree

The algorithms we have introduced thus far determine the topology of the tree corresponding to a given seed state and target momentum sector, but they do not determine the order in which the nodes of the tree are generated. For the numerics displayed in the previous sections we adopted the rule that, by default, we pause all branches and after each step of generating new descendents we find the highest weight state all of whose descendents we then generate. However, in Fig. 3d, e, f, we saw that this approach produces plateaus which indicate that less important states are generated before their more important counterparts. The main reason for this is that at a given node there are descendents with different levels of importance (as, for example, they have different numbers of particle-hole excitations). By generating all descendents of a node at the same time, states with more particle-hole pairs can be generated before states with fewer particle-hole pairs further down the tree. In this section we show how splitting the descendents of the momentum preserving leapwise scanning algorithm into three groups and treating them separately results in a more efficient algorithm.

The division of the descendents of a given node into three groups is done based on the number of additional particle-hole pairs they have compared to their parent, which is either zero, one, or two for the algorithms in this paper.[2] Given a node of the tree, we expect the states in the group of descendents with the same number of particle-hole pairs as the parent node to be of similar importance as the parent. Since we always consider the paused node with the highest weight, its descendents with the same number of particle-hole pairs are the ones we expect to be the most important unexplored eigenstates. Therefore we always start by generating these descendents if they have not yet been generated. In contrast to regular momentum preserving leapwise scanning, we do not also generate the descendents with additional particle-hole pairs at the same time.

Once we have generated the descendents with the same number of particle-hole pairs for a given node, we can generally not yet forget about this node as we could previously, since we still have to generate some of its descendents with additional particle-hole pairs. Therefore, we store it in a different list which tracks the states for which the descendents with no additional particle-hole pairs have been generated. Furthermore, we give it a weight by choosing a random descendent with an additional particle-hole pair and computing its weight, which is then used to weigh the parent node in this secondary list. Having populated both the initial list of paused nodes and this secondary list, we can choose to generate new descendents by either considering a state from the original list and generating more states with the same number of particle-hole pairs as their parent, or by considering a state from the secondary list, and generating the descendents of the node with one additional particle-hole pair. Which choice we make depends on where the state with the highest weight is, as it could be in the first list where its weight is that of the eigenstate, whereas in the second list it would be the weight of a randomly chosen descendent with an additional particle-hole pair.

In some cases, the full list of descendents would also have included states with two additional particle-hole pairs with respect to their parent node. In this case we follow a similar procedure and move the node from the secondary list to a third where we keep nodes whose descendents with zero and one additional particle-hole pair have been generated and associate to it another representative weight obtained by computing the weight of one of the states with

---

[2]This is due to the fact that the algorithms we consider change at most two quantum numbers per step of the algorithm. The approach outlined in this section can also be straightforwardly generalized to algorithms that can generate descendents with more than two additional particle-hole pairs by including additional groups.

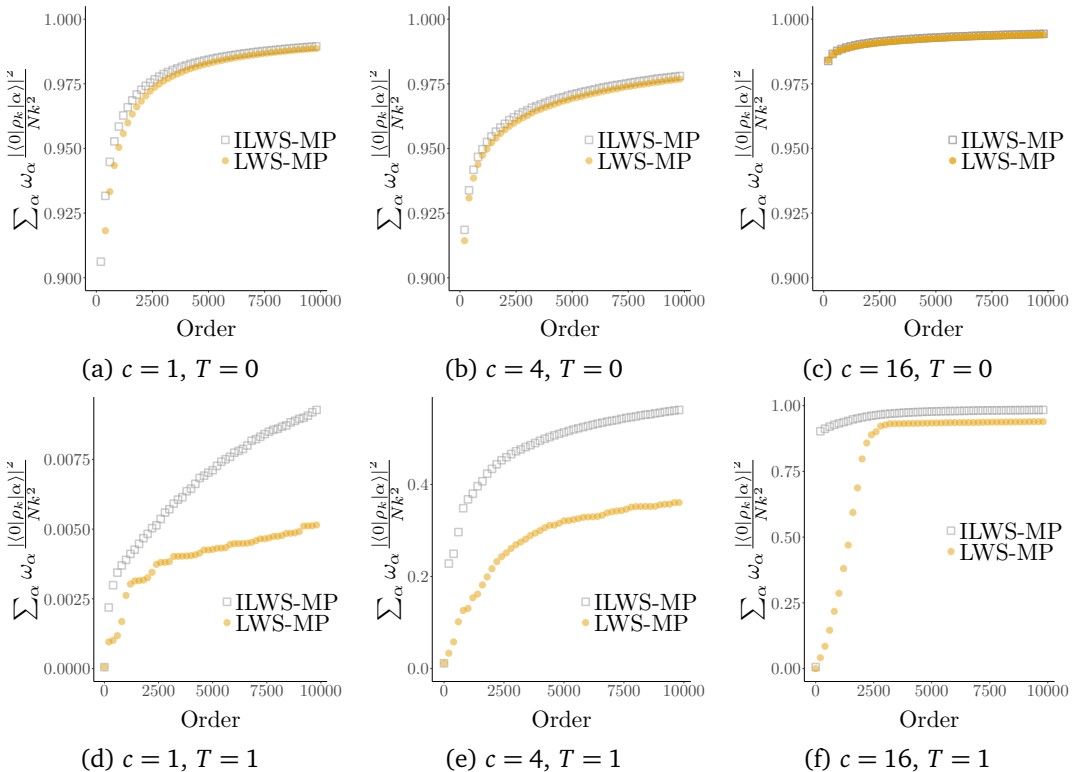

(a) $c = 1$, $T = 0$      (b) $c = 4$, $T = 0$      (c) $c = 16$, $T = 0$

(d) $c = 1$, $T = 1$      (e) $c = 4$, $T = 1$      (f) $c = 16$, $T = 1$

Figure 4: Comparison of the saturation of the $f$ sumrule with the number of states included in the summation between momentum preserving leapwise scanning (LWS-MP) and improved momentum preserving leapwise scanning (ILWS-MP). Starting from the ground state for $(a)$-$(c)$ and the representative thermal state at $T = 1$ for $(d)$-$(f)$, we generate 100,000 states for a target momentum of $k = \pi$, and $N = 128 = L$. We plot the sum rule saturation after every 200 states for $c = 1$ in $(a)$ and $(d)$, for $c = 4$ in $(b)$ and $(e)$, and for $c = 16$ in $(c)$ and $(f)$. Convergence at zero temperature sees a tiny improvement for improved momentum preserving leapwise scanning to regular momentum preserving leapwise scanning and remains near perfect for very few states. At finite temperature, we see a dramatic increase in performance in both overall convergence as well as the number of states required to achieve this convergence for improved momentum preserving leapwise scanning, eliminating all plateaus. Still, absolute convergence remains a challenge at lower values of the interaction strength for the number of states considered regardless of the algorithm used.

two additional particle-hole pairs. We then repeat the procedure outlined above with three lists. The resulting procedure is what we refer to as improved momentum preserving leapwise scanning (ILWS-MP).

The advantage of this approach is that, at the cost of computing at most two additional matrix elements per node, we can ensure that we are generating the states most important to the calculation under consideration. Despite this representing an additional computational effort, it can still represent a net gain as it can allow us to generate far fewer states for a given accuracy of the calculation under consideration. Whether the additional cost of computing these matrix elements is worth it can depend, for example, on the seed state under consideration. To illustrate this, consider the zero temperature dynamical structure factor calculation whose results are illustrated in Fig. 4a, b, c. Here we see that the performance the of regular

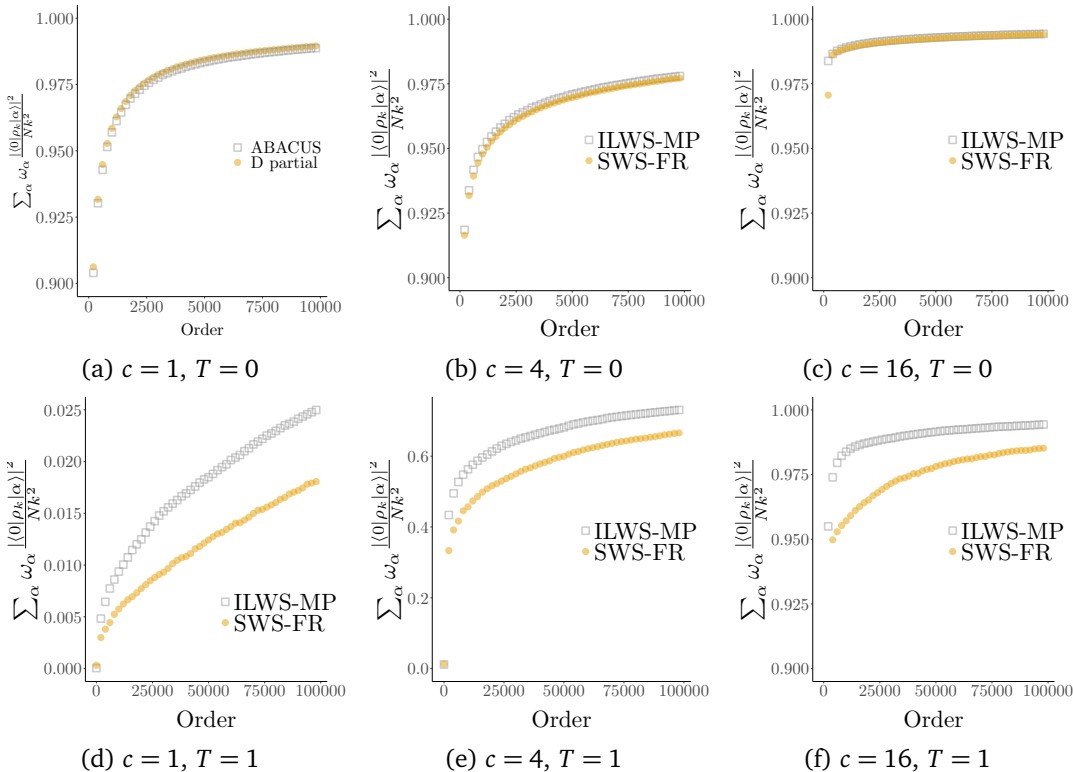

(a) $c = 1$, $T = 0$       (b) $c = 4$, $T = 0$       (c) $c = 16$, $T = 0$

(d) $c = 1$, $T = 1$       (e) $c = 4$, $T = 1$       (f) $c = 16$, $T = 1$

Figure 5: Comparison of the saturation of the $f$ sumrule with the number of states included in the summation between the improved momentum preserving leapwise scanning (ILWS-MP) and momentum preserving stepwise scanning with forced re-combinations. Starting from the ground state for $(a)$-$(c)$ and the representative thermal state at $T = 1$ for $(d)$-$(f)$, we generate 10,000 states for $(a)$-$(c)$ and 100,000 states for $(d)$-$(f)$ for a target momentum of $k = \pi$, and $N = 128 = L$. We plot the sum rule saturation after every 200 states for $c = 1$ in $(a)$ and $(d)$, for $c = 4$ in $(b)$ and $(e)$, and for $c = 16$ in $(c)$ and $(f)$. Convergence at zero temperature is virtually identical, whereas the improved momentum preserving scanning outperforms SWS-FR at finite temperature.

and improved momentum preserving leapwise scanning is virtually identical. On the other hand, when considering the finite temperature equivalent as illustrated in Fig. 4d, e, f, we see a big difference in performance. The algorithm where we allow some descendents to be generated at a later point in time outperforms the algorithm where all descendents are generated. Not only does it generate the same states quicker, but it also appears to achieve higher sum rule saturations in part because the algorithm where all descendents are generated gets stuck generating unimportant states.

# 8 Comparison to the state of the art

In this section, we compare the improved momentum preserving leapwise scanning algorithm to the most recent version of the state of the art software for the computation of correlation functions in the Lieb-Liniger model called ABACUS [1]. In Sec. 8.1 we consider the dynamical structure factor at zero and finite temperature. Since ABACUS was developed in part to compute the dynamical structure factor, it makes it the ideal candidate for a fair comparison. In Sec. 8.2

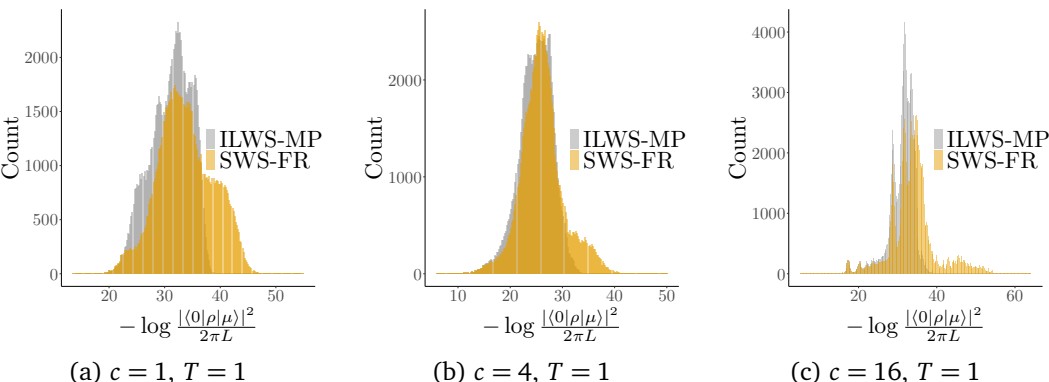

|(a) $c = 1$, $T = 1$|(b) $c = 4$, $T = 1$|(c) $c = 16$, $T = 1$|

Figure 6: Comparison of the histograms of the $f$ sumrule weights between the improved momentum preserving leapwise scanning (ILWS-MP) and SWS-FR. Starting from the representative thermal state at $T = 1$, we generate 10,000 states for a target momentum of $k = \pi$, and $N = 128 = L$. We plot the resulting histogram for $c = 1$ in (a), for $c = 4$ in (b), and for $c = 16$ in (c). We see that the improved momentum preserving leapwise scanning and SWS-FR find the same states with very large weights (those on the left), but SWS-FR generates more less important states (the states on the right).

we consider the generation of an optimal basis for the interaction quench. In both cases, we use commit 08C85CF590 of ABACUS as available at https://jscaux.org/git/jscaux/ABACUS. This version of ABACUS generates a tree topologically equivalent to the one generated by stepwise scanning, generating all its descendents by moving quantum numbers by at most one position. The difference between ABACUS and our momentum preserving stepwise scanning is the way the tree is built. Most crucially, ABACUS deals with the need to generate states with few particle-hole pairs by forcing the generation of branches of the tree where a particle and hole recombine. Therefore we can view it as stepwise scanning with forced recombinations, which we refer to with its abbreviation (SWS-FR) throughout this section. As we shall see, this is not the optimal way to deal with finite temperature states.

## 8.1   The dynamical structure factor

For the ground state dynamical structure factor calculation, illustrated in Fig. 5a, b, c, we see that the performance of SWS-FR and our improved momentum preserving leapwise scanning routine are virtually identical. Similarly to the discussion comparing momentum preserving stepwise and leapwise scanning, there is not a lot of freedom on how to generate the states when starting from the ground state. The differences which do exist are caused by states being generated in a different order due to a different way of building the same tree.

At finite temperature, we observe an increase in performance for the improved momentum preserving leapwise scanning compared to SWS-FR for all values of the interaction strength as illustrated in Fig. 5. At finite temperature, the seed states we consider no longer consist of a contiguous block of quantum numbers with empty positions between neighbouring quantum numbers being introduced. In this case, creating a new particle-hole pair leaves a hole that may not neighbour another quantum number that has not moved yet. For the improved momentum preserving leapwise scanning algorithm, states with the same number of particle-hole pairs can be direct descendents of this state by allowing the quantum number to move by more than one

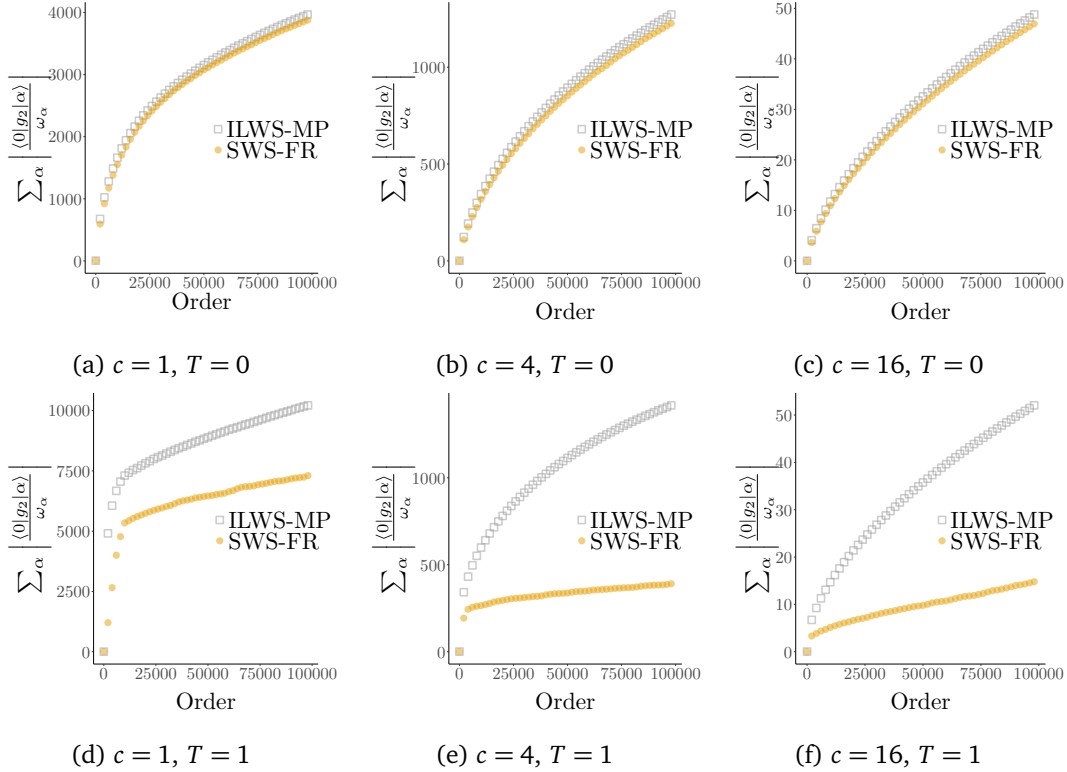

Figure 7: Comparison of the sum of eigenstate weights with the number of states included in the summation between the improved momentum preserving leapwise scanning (ILWS-MP) and momentum preserving stepwise scanning with forced recombinations (SWS-FR). Starting from the ground state for $(a)-(c)$ and the representative thermal state at $T = 1$ for $(d)-(f)$, we generate 10,000 states and 100,000 states respectively for a target momentum of $k = \pi$, and $N = 128 = L$. We plot the sum rule saturation after every 200 states for $c = 1$ in $(a)$ and $(d)$, for $c = 4$ in $(b)$ and $(e)$, and for $c = 16$ in $(c)$ and $(f)$. The differences between SWS-FR and the improved momentum preserving leapwise scanning for the sums at zero temperature are small whereas at finite temperature the latter outperforms SWS-FR by a large margin.

position. For example, we saw that we can have subtrees like

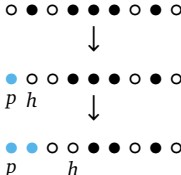

For the stewpwise scanning algorithm with forced recombinations such jumps are not allowed, forcing it to first create an additional particle-hole pair and move it to annihilate the initial

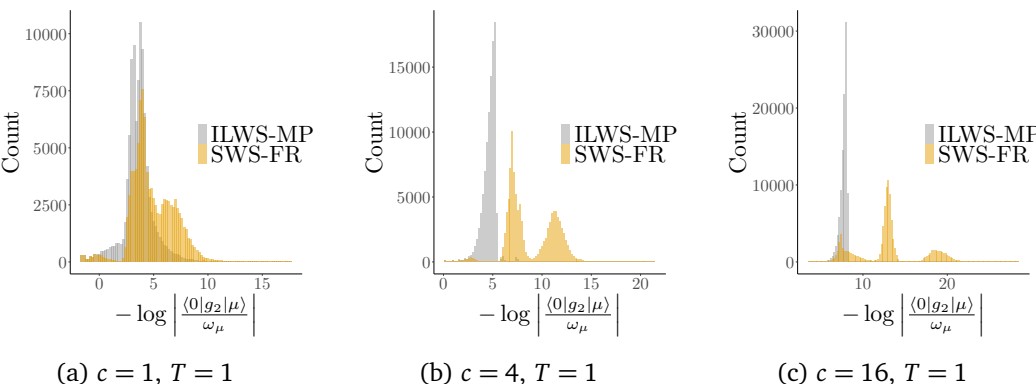

(a) $c = 1$, $T = 1$  (b) $c = 4$, $T = 1$  (c) $c = 16$, $T = 1$

Figure 8: Comparison of the histograms of the first 100,000 states generated by SWS-FR and the improved momentum preserving leapwise scanning (ILWS-MP) for the basis generation problem at $k = 0$ and $N = 128 = L$ at $T = 1$. We see that there is some overlap in which states are generated by both algorithms for the large weight states (on the left) but we also see that SWS-FR generates far more low weight states. At larger interaction strengths, we see that these lower weight states generated by SWS-FR are grouped in two distinct bumps.

hole leading to the following subtree:

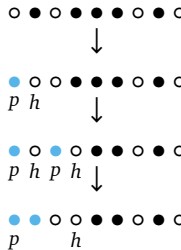

The difference may look innocent in this simple example, but as the distances between quantum numbers in the initial state grow and the number of such isolated quantum numbers increases, the number of additional states that have to be generated in order to reach all states with the same number of particle-hole pairs grows rapidly.

The fact that SWS-FR generates such less important intermediate states can be seen from Fig. 6. where we consider the histograms of the first 100,000 states generated by each algorithm. In these histograms, the $x$ axis measures the importance of an eigenstate for the saturation of the f-sum rule, where the importance decreases as $x$ increases. The rightmost gray part of the histograms represent the intermediate states that are generated too soon due to the sub-optimal topology of the tree that SWS-FR is building. Since this problem is related to the topology of the tree, it occurs independently of the interaction strength considered.

## 8.2 Generating a basis for the interaction quench

Another way to benchmark our algorithm is by considering the problem of a quench in the interaction strength from $c_i$ to $c_f$ [54]. In this case, we want to find an approximate expansion of some initial eigenstate $|\Psi_0\rangle$ of $H(c_i)$ in terms of eigenstates of $H(c_f)$. Truncated spectrum methods can be used to obtain the expansion coefficients $b_\alpha$ in

$$|\Psi_0(t)\rangle = \sum_\alpha b_\alpha e^{-iE_\alpha t} |\alpha\rangle. \tag{18}$$

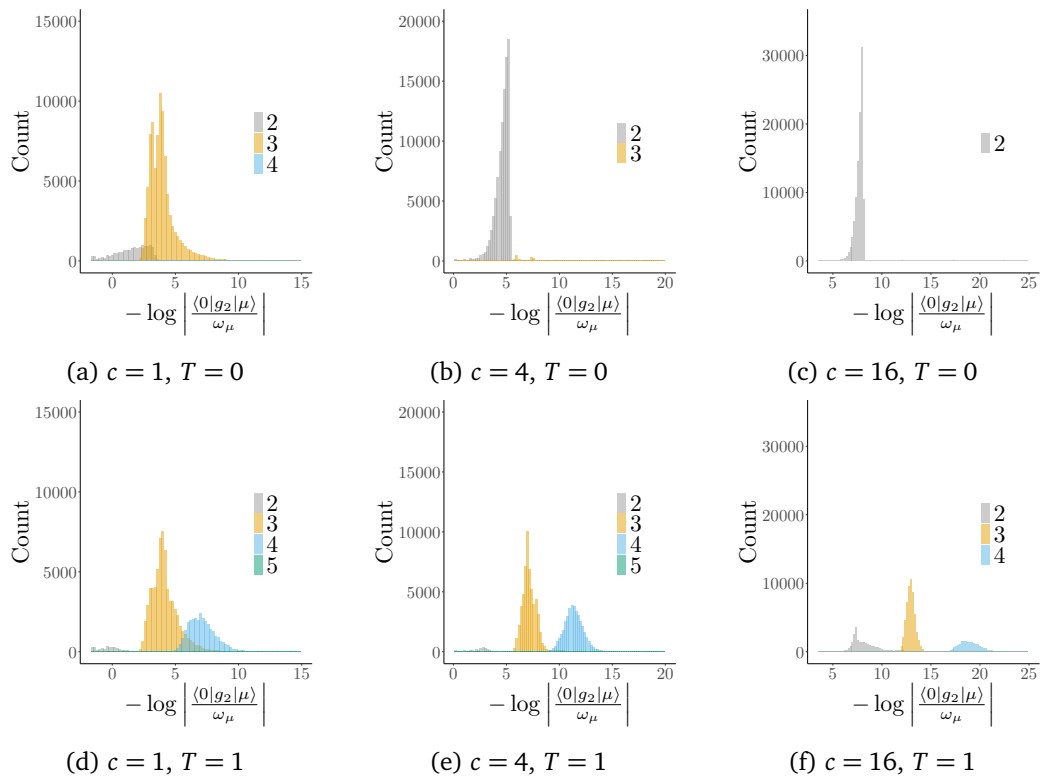

Figure 9: Breakdown of the histograms shown in Fig. 8 based on the number of particle-hole pairs for improved momentum preserving leapwise scanning (ILWS-MP) (a)-(c) and SWS-FR (d)-(f). We see that the bumps of unimportant states generated by SWS-FR are states from the three and four particle-hole sectors whereas improved momentum preserving leapwise scanning is capable of sticking to lower particle-hole sectors.

In order to obtain an expansion that captures the time evolution following the quench accurately, we need to choose our basis states $|\alpha\rangle$ wisely. In [54] we showed that a good estimate of the importance of an eigenstate $|\alpha\rangle$ is given by

$$w(|\alpha\rangle) = \left| \frac{\langle \Psi_0 | g_2 | \alpha \rangle}{E_{\Psi_0} - E_\alpha} \right| . \tag{19}$$

The task of our scanning algorithms is therefore to generate the eigenstates with the largest weights.

In order to compare our algorithm to the momentum preserving stepwise scanning algorithm with recombinations, we compare the sums of the weights generated having generated $i$ eigenstates as shown in Fig. 7. We see that at zero temperature, the results from the improved momentum preserving leapwise scanning are only marginally better than those of SWS-FR, which can again be explained by the fact that at zero temperature the topology of trees generated by both algorithms is identical. At finite temperature however, we see a much more pronounced difference than we saw for the finite temperature calculation of the dynamical structure factor.

So why is the difference between SWS-FR and the improved momentum preserving leapwise scanning for this basis generation problem so much more pronounced compared to the finite temperature dynamical structure factor calculation? Note that the $f$ sum rule for the dynamical structure factor is dominated by the one particle-hole sector, which both algorithms are

perfectly capable of generating in full, as it contains at most $N$ states at fixed momentum. For the basis generation problem considered here however, the states contributing most strongly are those in the two particle-hole sector (states in the one particle-hole sector cannot have the same momentum as the seed state). The process of generating a given state in the two particle-hole sector is however very different between the improved momentum preserving leapwise scanning and SWS-FR. As we mentioned, SWS-FR is forced to generate intermediate states, of which there are increasingly many as temperature increases. The improved momentum preserving leapwise scanning on the other hand allows for jumps of quantum numbers by more than one position allowing it to generate these states without intermediate states resulting in a more efficient calculation. The choice of topology of the tree combined with the versatile way of building the tree is thus what allows the algorithm to focus on the contributions that are most important at a given point in the calculation, as illustrated by the histograms of contributions in Fig. 8

In order to substantiate our claim that our algorithm outperforms SWS-FR due to the different topology of the tree, consider Fig. 9. Here we see a breakdown of the histograms of weights generated by either the improved momentum preserving leapwise scanning ($a$) through ($c$) or SWS-FR ($d$) through ($f$) based on the number of particle-hole sectors the states whose weights are displayed are from previously shown in Fig. 8 based on the number of particle-hole pairs of a given contribution. We see that SWS-FR spends its time generating a lot of unimportant states from the three and four particle-hole sectors in order to generate the important contributions from the two particle-hole sector. Since for the improved momentum preserving leapwise scanning any state can be generated without intermediate states with more particle-hole pairs, it is able to avoid such problems. We therefore conclude that the improved momentum preserving leapwise scanning algorithm is better suited for dealing with problems at finite temperature than algorithms that are not able to strictly preserved the number of particle-hole pairs like stepwise scanning or SWS-FR. We have seen that this is especially true if the calculation under consideration is not one dominated by the one particle-hole sector.

## 9 Conclusions

Despite the powerful analytical tools that have been developed for integrable systems, performing a numerical evaluation over eigenstates is still often an inevitable step required to compute correlation functions. It is therefore crucial that we have good Hilbert space exploration algorithms that allow us to accurately and efficiently approximate such summations. Here we have reviewed the basic principles that such an algorithm has to satisfy and developed a number of concrete examples that satisfy these criteria. Starting from the most basic algorithm satisfying these basic principles, we considered its shortcomings one by one and proposed solutions resulting in incremental changes that led us to the final algorithm. Finally, we compared this algorithm to the state of the art for the dynamical structure factor at zero and finite temperature as well as the problem of generating an efficient basis for a quench in the interaction strength.

Overall, our algorithms can be viewed as an algorithm for building a single-rooted tree where every node of the tree represents an eigenstate and the algorithm specifies the topology of the tree. Since the tree is infinite and we only have finite computational resources it is also important in what order the nodes of the tree are generated as this determines what the tree looks like after some finite time. A key characteristic for the importance of a state is the number of particle-hole pairs it has with respect to the seed state. In order to avoid having to consider less relevant states with more particle-hole pairs we chose our final algorithm to

have a topology where any node with $n$ particle-hole pairs can be generated without having to generate nodes with more particle-hole pairs. This topology, in combination with a clever way of choosing the order in which to generate new states is what led to the most efficient algorithm.

Comparing our final algorithm to the current version of the state of the art library ABACUS showed that we outperform the latter when considering the dynamical structure factor at finite temperature as well as the problem of generating an efficient basis for a quench in the interaction strength at finite temperature. In the latter case the difference is particularly pronounced, emphasizing the importance of the choice of topology. After all, the reason that our algorithm outperforms ABACUS is primarily due to the fact that if ABACUS wants to generate certain states with $n$ particle-hole pairs it has to go through states with more particle-hole pairs. The examples considered show that the improved momentum preserving leapwise scanning algorithm offers significant advantages for finite temperature calculations. One of the main advantages is due to the topology of the tree being generated, which allows any state from the $n$ particle-hole sector to be generated without generating states from higher particle-hole sectors. This is especially important when considering calculations not dominated by the one particle-hole sector as the number of intermediate states required to construct a state from the $n$ particle-hole sector grows dramatically with increasing $n$ at finite temperature. Improved momentum preserving leapwise scanning is therefore more suited to dealing with problems involving for example the $g_2$ and $g_3$ operators, the latter being relevant to modelling three-body losses. Because of these advantages, future versions of ABACUS will incorporate this approach to be better equiped to deal with finite temperature calculations.

In this work we considered calculations where the number of particles of the states we explore is equal to the number of particles of the reference state. However, for some problems this is not the case (e.g. when calculating the Green's function). In these cases additional difficulties arise for finite temperature states since in this case the number of particle-hole pairs is no longer well-defined. Future research is required to come up with strategies to efficiently deal with these problems. Another interesting direction for future research direction would be the extension of the ideas in this paper to spin chains. Despite their differences, such as the existence of string solutions, the spin chain and the Bose gas also share key characteristics. For example, the quantum numbers of the spin chain can be viewed as a multi-level version of those of the Bose gas, each lattice representing the quantum numbers of a given string sector which can be visualized as:

$$\vdots$$

$$2\text{-strings:} \quad \cdots \circ \circ \circ \circ \bullet \circ \bullet \circ \circ \cdots$$

$$1\text{-strings:} \quad \cdots \circ \circ \bullet \bullet \bullet \circ \bullet \circ \cdots$$

An algorithm like the one we developed can then be applied to the quantum numbers of every string sector. However, additional complications due to the differences are bound to arise. For example, in the spin chain more care has to be taken to avoid overcounting since a given eigenstate can be represented by multiple quantum number configurations. Furthermore, additional constraints such as the magnetization, as well as the periodicity of the momentum will require careful consideration.

## Acknowledgements

We are grateful to Neil Robinson for his valuable insights and his suggestions during the preparation of this manuscript.



## Funding information

This work received funding from the European Research Council under ERC Advanced grant No 743032 DYNAMINT.

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
