# Peer review of "Improved Hilbert space exploration algorithms for finite temperature calculations"

_SciPost Physics Core, doi:SciPost Phys. Core 6, 039 (2023)_

## Round 1 · Referee Report · Anonymous (Referee 1) · 2023-2-28

Strengths

1) important for quantitative and precise predictions for dynamic correlation functions and non-equilibrium quench dynamics of integrable models.

2) throughout considerations of Hilbert space exploration algorithms for particle-hole excitations

3) the aims, issues and solutions very well explained

Weaknesses

I can't really see any.

Report

In this paper authors considers various algorithms for exploring Hilbert space of the Lieb-Liniger model constrained by fixed number of particles. Understanding relevant states (excitations) is important for computation of finite-temperature dynamic correlation functions or generating a basis for post-quench state. The paper presents essentially efficiency analysis of different algorithms and the main result is that the new algorithm developed by the authors surpasses the ABACUS algorithm (developed by one of the authors) which is successfully used over the almost last two decades. The paper is quite technical which I find perfectly suited for the scope of this journal.

My recommendation is to publish the paper. Below I list some minor comments.

Requested changes

1) I think the authors assume that the sets of quantum numbers are always ordered, however I didn't find such statement in the text.

2) On pg. 9, when visualising the right and left hopping particles, two sets of quantum numbers are slightly misaligned. Is there a meaning to this relative shift?

3) In the last paragraph of pg. 18 the authors write that the descendants are grouped based on the number of extra particle-hole pairs and that this number can be either zero, one or two. I don't quite see why this number couldn't be larger. Is this a restriction put by hand?

4) The caption of fig. 5 states the number of states generated to be 10 000 while the bottom figures seems to go up to 100 000 states.

Some typos: 1) at the bottom of pg. 7, 'would not' -> 'were not' , 2) at the top of pg. 18, extra 'the' before practically, 3) right before conclusions on pg. 25, 'domtinated'.

  • validity: -
  • significance: -
  • originality: -
  • clarity: -
  • formatting: -
  • grammar: -

Author:  Albertus de Klerk  on 2023-04-04  [id 3545]

(in reply to Report 1 on 2023-02-28)

We thank the referee for their careful reading, and agree with the summary of the paper. Furthermore, we thank the referee for their support for our work and their suggestions to improve the paper. In the following we go into the changes requested by the referee.

1) The referee is indeed correct when saying that we assume the sets of quantum numbers to be ordered. To clarify this point, we have added a statement to this extent in the first sentence of section 4 where we first talk about visualizing the quantum numbers.

2) We commend the referee on their keen eye. This relative shift was not intentional and bears no meaning, but was instead a typesetting artefact created by the arrow in the same figure. As such, the relative shift has been removed.

3) The fact that descendents of a given parent state can have at most two additional particle-hole pairs is a consequence of our choice to generate descendents by changing at most two quantum numbers. To clarify that this statement is specific to the algorithms we consider in our paper we have changed the sentence to reflect this and added a footnote discussing this further.

4) The referee is indeed correct in noting that the bottom row of Figure 5 containing subfigures (d)-(f) concerns 100,000 states whereas the top row with subfigures (a)-(c) show a 10,000 states. We have corrected the caption accordingly.

The typos have also been corrected, and we thank the referee again for their careful reading.

---

## Round 1 · Referee Report · Anonymous (Referee 2) · 2023-3-9

Report

In this paper, the authors consider a technical but very important problem in the theory of many-body quantum integrable systems. Namely, the problem is to find an efficient scheme to evaluate numerically spectral sums appearing in the computation of correlation functions. Although eigenstates and matrix elements (form factors) of local operators are known analytically in many cases, often this evaluation step can not be avoided, making this problem urgent.

ABACUS is an algorithm developed by one of the authors which gives us the state-of-the-art performance to tackle this problem. In this paper, the authors put forward a modification of the algorithm outperforming ABACUS in states of large entropy. Because these appear naturally in finite-temperature and out-of-equilibrium problems, this is a very important result. The authors provide full numerical evidence supporting the claims.

The paper is necessarily technical, but I think it is written well and is very clear. I also checked the list of references which in my opinion is fair and complete. Regarding the results, I am sure this new approach will be very useful in future applications of the improved ABACUS algorithm, so it will have a significant impact.

I don't have comments on how this paper could be improved. Given the importance of the result, I therefore recommend publication of the draft as is.
  • validity: -
  • significance: -
  • originality: -
  • clarity: -
  • formatting: -
  • grammar: -

Author:  Albertus de Klerk  on 2023-04-04  [id 3546]

(in reply to Report 2 on 2023-03-09)

We thank the referee for reading and reviewing our paper, and agree with their summary of it. Furthermore, we thank the referee for their support for our work and their recommendation to publish our paper.

---

## Editorial Decision

published